# Designing BERT for Convolutional Networks: Sparse and Hierarchical Masked Modeling

**Keyu Tian**[1,2,3], **Yi Jiang**[2*], **Qishuai Diao**[2],
**Chen Lin**[4], **Liwei Wang**[1*], **Zehuan Yuan**[2]
[1]Center for Data Science, Peking University    [2]Bytedance Inc
[3]Pazhou Lab (Huangpu)    [4]University of Oxford
keyutian@stu.pku.edu.cn, {jiangyi.enjoy,diaoqishuai}@bytedance.com,
chen.lin@eng.ox.ac.uk, wanglw@pku.edu.cn, yuanzehuan@bytedance.com

## Abstract

We identify and overcome two key obstacles in extending the success of BERT-style pre-training, or masked image modeling, to convolutional networks (convnets): (i) convolution operation cannot handle irregular, randomly masked input images; (ii) the single-scale nature of BERT pre-training is inconsistent with convnet's hierarchical structure. For (i), we treat unmasked pixels as sparse voxels of 3D point clouds and use sparse convolution to encode. This is the first use of sparse convolution for 2D masked modeling. For (ii), we develop a hierarchical decoder to reconstruct images from multi-scale encoded features. Our method, called *Spar*se mas*K*ed modeling (*SparK*), is general: it can be used directly on any convolutional model without backbone modifications. We validate it on both classical (ResNet) and modern (ConvNeXt) models: on three downstream tasks, it surpasses both state-of-the-art contrastive learning and transformer-based masked modeling by similarly large margins (around $+1.0\%$). The improvements on object detection and instance segmentation are more significant (up to $+3.5\%$), validating the strong transferability of features learned. We also find SparK's favorable scaling behavior by observing more gains on larger networks. All of these findings support the promising future of generative pre-training on convnets. Both codes and pre-trained models have been released at https://github.com/keyu-tian/SparK.

## 1 Introduction

The pretrain-finetune paradigm in natural language processing (NLP), as exemplified by BERT and GPT (Devlin et al., 2018; Clark et al., 2020; Radford et al., 2019; Brown et al., 2020), is remarkably effective and thus long envied by our vision community. It is the emerging masked image modeling (Bao et al., 2021; He et al., 2021; Xie et al., 2021; Chen et al., 2022) initially extends the success of BERT *from language transformers to vision transformers* (ViTs). A bold move that increases the mask ratio to a staggering level (60~75%) is largely credited with this success (He et al., 2021; Xie et al., 2021). As a result, the field of visual self-supervised learning on ViTs (Dosovitskiy et al., 2020; Liu et al., 2021) has now shifted from contrastive learning (Grill et al., 2020; Chen et al., 2021; Caron et al., 2021) to BERT-style masked modeling or a fusion of the two (Zhou et al., 2021).

Despite this progress, extending the success of BERT pre-training *from transformers to convolutional networks* (convnets) remains a desirable but unrealized goal. Early pioneering work (Pathak et al., 2016) predated BERT but performed much worse than supervised pre-training. Although there have been efforts over the past year to port BERT to convnets, they ultimately compromise by proposing a non-convolutional model (Gao et al., 2022) or non-masked modeling (Fang et al., 2022). One might therefore wonder: *what exactly is impeding the application of BERT to convnets?*

We try to conclude that in essence, the difficulty is rooted in the fundamental differences in data processing between language and vision (Bateman, 2014; Cheng et al., 2022). While typical NLP models like recurrent networks or transformers process text as a variable-length sequence of words (well-defined semantic units), convnets have to recognize objects of different sizes from raw pixels ("units" at different scales). This large disparity rises two challenges: **(i) Removing the information**

---

[*]Corresponding authors: jiangyi.enjoy@bytedance.com, wanglw@pku.edu.cn

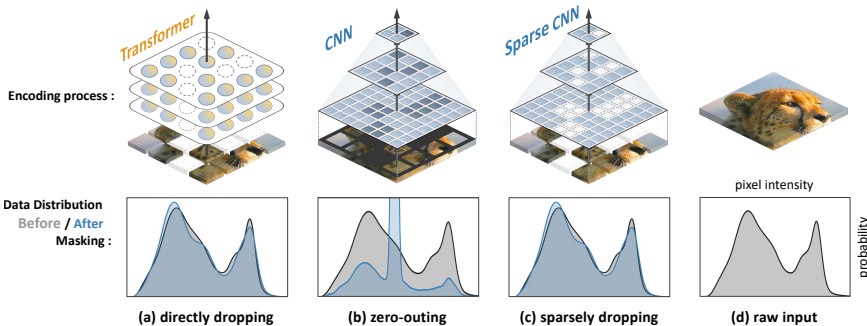

Figure 1: **Different masking strategies** with pixel intensity histograms plotted before (in gray) and after (blue) masking. (b) is a straightforward idea to apply masked modeling to convnets, which results in a distribution shift. (a) illustrates MAE (He et al., 2021) that has no such side effect thanks to the transformer's ability to process variable-length input. We propose (c) to adapt convnets to irregular masked input without a distribution shift.

**of masked "words" is difficult for convnets.** In ViTs, an input image is divided into non-overlapping patches. Simply dropping masked patches or replacing them with mask tokens can remove the information. This ease relies on transformer being able to handle *irregular (variable-length)* and *non-overlapping* patches, thus cannot be achieved on convnets as they not only operate on *regular* grids, but also perform sliding window *with overlapping*. One may zero-out all masked pixels and feed this "mosaic" into a convnet, but this would lead to a significant distribution shift (in figure 1) and other issues (discussed further in section 3.1 and figure 3), thus cannot be an ideal solution. **(ii) Single-scale algorithms are inadequate for learning multi-scale (hierarchical) features.** Multi-scale structures have been a gold standard in computer vision, which allows visual processing systems like SIFT descriptors (Lowe, 1999; Bay et al., 2006) and pyramid networks (He et al., 2015; Lin et al., 2017) to handle variations in object scale. In contrast, the masked modeling approach from NLP operates in a single-scale manner. Applying it directly on convnets will miss the advantage of model hierarchy.

In this work, we clear the hurdles above and make BERT suitable for convnet by proposing **Spar**se mas**K**ed modeling with hierarchy (SparK). We first randomly mask an image in a patch-wise manner. Observing the sparse nature of point clouds coincides with these unmasked patches, we treat them as a flatten point cloud and use sparse convolution for encoding. This enables convnets to handle irregular masked images. For decoding and reconstruction, the sparse features are filled with mask embeddings and fed into a multi-scale decoder, leveraging the hierarchical structure of convnets.

SparK is a general method that does not limit the specific encoder to be pre-trained. We test it with two representative convnet families: classical ResNets (He et al., 2016) and modern ConvNeXts (Liu et al., 2022). All models benefit from SparK, with more gains on larger models that demonstrates its favorable scaling ability. On standard downstream tasks (classification, object detection and instance segmentation), convnet-based SparK outperforms both (i) state-of-the-art contrastive learning and (ii) transformer-based masked modeling by similarly large margins (around $+1.0\%$). The improvements over COCO baselines are more significant than those on ImageNet (up to $+3.5\%$), indicating the representations learned by SparK are highly transferable. To summarize, SparK provides:

- The first pre-training method in the style of BERT that can be directly applied to any convnets without backbone modifications, overcoming their inability to handle irregular masked inputs.

- The insights into the design of generative pre-training for convnets, *e.g.*, the first use of sparse convolution for masked image modeling and a hierarchical design for BERT-style pre-training.

- A leap in convnet's performance across downstream tasks with gains of up to 3.5 points, showing the promise of extending the success of transformer's pretrain-finetune paradigm to convnets.

The recent surge of interest in vision transformers (Liu et al., 2021; He et al., 2021) has shifted the focus away from convnets in the computer vision community. However, convnets embody the core principles of many classical vision processing systems, such as scale- and translation-equivariance, locality, weight-sharing, and hardware-friendliness (Lowe, 1999; Csurka et al., 2004). These networks continue to be indispensable in addressing a variety of challenging and structural real-world tasks beyond classification (Jaderberg et al., 2015; Liu et al., 2017; 2022). We hope SparK's inspiring performance will prompt us to revisit convnets as generic backbones for computer vision community, and motivate more future arts in exploiting their potential through generative pre-training.

## 2 RELATED WORK

### 2.1 HIERARCHICAL VISUAL PROCESSING SYSTEMS

**Hierarchical structure** is acknowledged as a gold standard for visual representation systems. Many fundamental handcrafted feature descriptors (Lowe, 1999; Bay et al., 2006; Rublee et al., 2011) extract multi-scale visual representations via scale-space extremum on feature pyramid (say, octave). The crux behind this hierarchical design is to extract scale-invariant (or equivariant) features, thus, allows the system to cope with varying object sizes (scales). Widely used in visual tasks (Felzenszwalb et al., 2008; Yang et al., 2009), these descriptors also motivate the design principles of convolutional networks (He et al., 2016; Tan & Le, 2019; Liu et al., 2022). Some recent arts also elaborately design hierarchical modules that allow the information aggregation at different granularities to better tackle detection and segmentation tasks using convnets (Long et al., 2015; Liu et al., 2016; Lin et al., 2017).

### 2.2 RECENT PROGRESS ON VISUAL SELF-SUPERVISED LEARNING

**Recently, the contrastive learning** formulates self-supervise learning as an instance classification task (Van den Oord et al., 2018; He et al., 2020; Chen et al., 2020a). Efforts have been made (Grill et al., 2020; Caron et al., 2020; Chen & He, 2021) to overcome the core issue of mode collapse. More advanced methods are developed since then (Tian et al., 2020; Zbontar et al., 2021; Li et al., 2023; Chen et al., 2021), and this line of work had dominated the area of visual unsupervised learning until masked generative pre-training along with the vision transformer architecture came into view.

**Masked image modeling,** inspired by the recent success of masked language modeling in natural language processing (NLP) (Devlin et al., 2018; Liu et al., 2019), has attracted growing interest for visual pre-training. The pioneering work (Bao et al., 2021) pre-trains vision transformers by learning to predict token indices of masked patches. He et al. (2021) takes advantage of transformer's ability to handle variable-length inputs and implements an efficient and scalable method. Both He et al. (2021) and Xie et al. (2021) regress raw RGBs to simplify the pre-training, while Wei et al. (2022) and Li et al. (2022) selects HOG (Dalal & Triggs, 2005) or frequencies as targets due to their rich semantics or structures. Gao et al. (2022) designs a transformer with a heavier patchifier to perform masked modeling. Zhang et al. (2022) have verified this idea in 3D computer vision. So far, enormous studies have successfully verified the efficacy of these algorithms on vision transformers (Zhou et al., 2021; Chen et al., 2022). However, on the other hand, their methodology is almost the same as that in NLP (Devlin et al., 2018; Liu et al., 2019), and is therefore difficult to be used for hierarchical convolutional models – on convnets, contrastive learning still remains state-of-the-art.

### 2.3 SPARSE CONVOLUTION FOR VISUAL REPRESENTATION

Convolution is widely used in 2D computer vision (Dalal & Triggs, 2005; He et al., 2016), which typically performs sliding window on regular grids (pixels). When facing with 3D point clouds, this operator quickly becomes unaffordable due to the cubic increasing number of grids (voxels). Considering point clouds are highly sparse and irregular, one can skip all empty voxels for speed. This motivates the **sparse convolution** (sparseconv) (Liu et al., 2015), which is heavily used in modern convnets for 3D visual tasks (Zhou & Tuzel, 2018; Sindagi et al., 2019). Minkowski Engine (Choy et al., 2019) is one of the most common sparseconv frameworks. Some prior arts (Verelst & Tuytelaars, 2020) also tried to introduce sparseconv for faster 2D visual understanding. And in this work, we have observed the similarity between 3D point clouds and 2D masked images in BERT-style pre-training. We thus use sparseconv, for the first time, with the purpose of "facilitating the adaptation of convnet to BERT masked modeling", rather than of "speeding up the computation of convolution".

## 3 APPROACH

Illustrated in figure 2, our SparK framework aims to pre-train a convolutional network encoder via hierarchical masked image modeling – masking a portion of image and learning to recover it. We are going to detail SparK by introducing a sparse masking strategy (section 3.1), a hierarchical encoder-decoder architecture (section 3.2), and the optimization target of SparK pre-training (section 3.3).

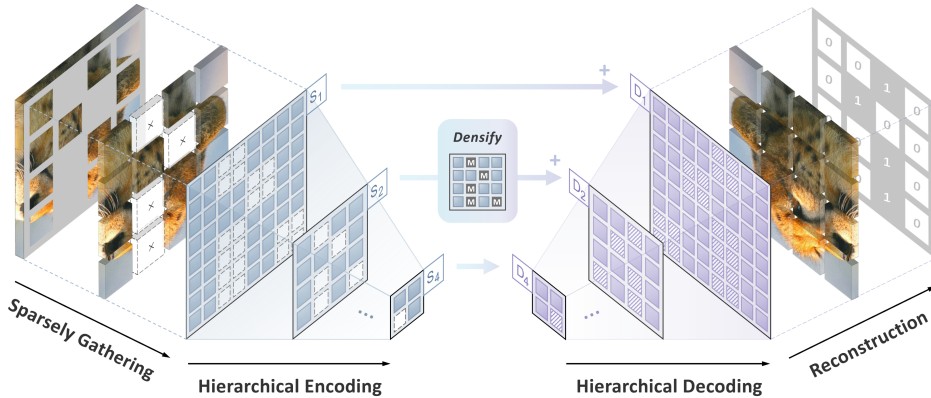

Figure 2: **Sparse masked modeling with hierarchy.** To adapt convolution to irregular masked input, visible patches are gathered into a sparse image and encoded by sparse convolution. To pre-train a hierarchical encoder, we employ a UNet-style architecture to decode multi-scale sparse feature maps, where all empty positions are filled with mask embedding. This "densifying" is necessary to reconstruct a dense image. Only the regression loss on masked patches will be optimized. After pre-training, only the encoder is used for downstream tasks.

### 3.1 SPARSELY GATHERING UNMASKED PATCHES

We start by the patch-wise masking strategy widely used in masked image modeling. An image is divided into several non-overlapping square patches, each of which will then be masked independently with a given probability called mask ratio. The key to a masked image modeling algorithm is how to eliminate the pixel information from these masked patches.

**Previous transformer-based masked modeling** can easily eliminate the information by directly removing masked patches or replacing them with a mask token. This ease relies on the fact that vision transformers are born to handle *irregular* (variable-length) input and operate on *non-overlapping* image patches. Since convnets cannot do this, new approaches have to be sought. A straightforward idea is to set all masked pixels to zero and feed this image to a convnet. This, however, has three evident shortcomings: (i) the computation on masked regions is redundant; (ii) it would disturb the data distribution of pixel values, as illustrated in figure 1; (iii) the patterns on mask maps will vanish after applying several convolutions to this zero-out masked image. We examine problem (iii) in figure 3, where we also give our solution. Note that this problem is particularly acute when using modern deep convnets due to the large number of successive convolutional blocks.

**To overcome the problems, we propose to sparsely gather all unmasked patches** into a sparse image, and then use **sparse convolutions**[1] to encode it. This strategy: (i) ensures no information is leaked; (ii) can be applied directly to any convnet without backbone modifications; (iii) is efficient as sparse convolution computes only at visible places; (iv) solves the aforementioned issues of "pixel distribution shift" and "mask pattern vanishing". As shown in figure 3, sparse convolution will skip all masked positions on sparse feature maps, and only computes at unmasked points. This helps to prevent the shape of the mask pattern from changing with convolution, thus ensures a consistent masking effect and ratio throughout all convolution layers. Another fact is that when fine-tuning, all sparse convolutional layers can be naturally reduced to ordinary dense ones. This is true because dense images are actually the special cases of sparse images that have no "holes".

### 3.2 HIERARCHICAL ENCODING AND DECODING

**By "hierarchical" encoding,** we mean the encoder will generate a set of feature maps with different resolutions, namely different *scales*. Taking a ResNet-style model for example, it typically contains 4 stages each with a series of convolutional blocks and a downsampling module. The feature resolution is downsampled by a factor of 2 after every stage. For an image shaped as $H \times W$, a ResNet-50 produces feature maps at 4 scales with resolutions of $\frac{H}{4} \times \frac{W}{4}$, $\frac{H}{8} \times \frac{W}{8}$, $\frac{H}{16} \times \frac{W}{16}$, and $\frac{H}{32} \times \frac{W}{32}$. Let $S_1$, $S_2$, $S_3$, and $S_4$ be these sparse features, respectively. They will be used to decode.

---

[1]By "sparse convolution", we mean the *submanifold sparse convolution* that computes only when the kernel center covers a non-empty element. Please refer to Graham & van der Maaten (2017) for more details.

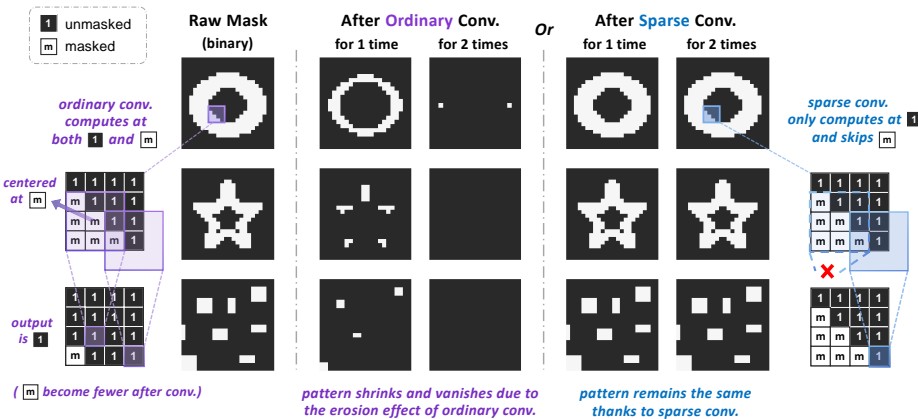

Figure 3: **Using sparse convolution to address "mask pattern vanishing" issue.** Three mask examples are shown. As in left, when computing ordinary "dense" convolution centered at a zero (masked) position, the result would be non-zero if the filter covers any non-zero (unmasked) points. Repeating this convolution will *erode* masked regions (zero positions) and *dilate* the unmasked ones, leading to the vanishing problem. We use sparse convolution to overcome this undesired property by skipping all masked positions and keeping the mask pattern.

**Overall, the decoder** follows the design of UNet (Ronneberger et al., 2015). We use a relatively light decoder that contains three successive blocks $\{\mathcal{B}_3, \mathcal{B}_2, \mathcal{B}_1\}$ with upsampling layers. Before reconstructing a dense image, it is necessary to *fill in* all the empty positions on sparse feature maps. This is called "**densifying**". Taking the smallest sparse feature $S_4$ as example, all empty positions (inactive sites) on $S_4$ are filled with a mask embedding [M$_4$] to get a dense feature $S_4'$. A projection layer $\phi_4$ is applied then, in case encoder and decoder have different network widths:

$$D_4 = \phi_4(S_4'). \tag{1}$$

So $D_4$ is the input of decoder's first block $\mathcal{B}_3$. It has the same resolution of $S_4$ with $\frac{H}{32} \times \frac{W}{32}$. Similarly, we can get $D_3$, $D_2$, and $D_1$ (with shapes of $\frac{H}{16} \times \frac{W}{16}$, $\frac{H}{8} \times \frac{W}{8}$, $\frac{H}{4} \times \frac{W}{4}$) via:

$$D_i = \mathcal{B}_i(D_{i+1}) + \phi_i(S_i') \qquad (\forall\, i \in \{3, 2, 1\}). \tag{2}$$

Note that four different mask embeddings [M$_{4\sim1}$] and projection layers $\phi_{4\sim1}$ are required: they belong to different scales, and may have different network widths. The final output of decoder is $D_1$.

### 3.3 Optimization Target and Transferring to Downstream

**To reconstruct an image from $D_1$,** a head module $h$ is needed, which should include two more upsampling layers to reach the original resolution of input $H \times W$. As for the reconstruction target, we choose per-patch normalized pixels as targets with an $L^2$-loss, and calculate errors only on masked positions. These designs have been proven to facilitate models to learn more informative features in He et al. (2021), and are also verified by the ablation study later in section 4.5.

**After pre-training,** we discard the decoder and only use the encoder for downstream tasks. When fine-tuning, the pre-trained sparse encoder can be directly generalized to dense images without any tuning, due to the fact that dense input is a special case of the sparse, where every position is active.

## 4 Empirical Results

### 4.1 Implementation Details

**Components.** SparK can use any convolutional network as the encoder, without any special design of the backbone architecture. We implement SparK with two of the most representative convnet families: classical ResNets (He et al., 2016) and modern ConvNeXts (Liu et al., 2022). One can easily test SparK on other convolutional architectures as well. As for the mask embeddings [M$_{4\sim1}$], we implement them as random-initialized learnable feature vectors. For decoding, we use a lightweight UNet decoder. See Appendix A for its detailed structure. Positional embeddings are not used since convnet already encodes the spatial information. We also test this in the ablation study (section 4.5).

**Simple implementation of pre-training.** For simplicity the same hyperparameters are used for all architectures (ResNets, ConvNeXts) and model sizes, even though tuning each may improve our fine-tuning performance at face value. All models are pre-trained with 1.28 million unlabeled images from ImageNet-1K (Deng et al., 2009) training set for 1600 epochs. Only the minimal augmentation is required (random cropping and horizontal flipping). We use the same mask patch size (32) and ratio (60%) as in SimMIM (Xie et al., 2021). We train with a LAMB optimizer (You et al., 2019), a batch size of 4096, and a cosine-annealing learning rate with peak value $= 0.0002 \times batchsize/256$.

**Fine-tuning.** We use the official implementations of ResNet (Wightman et al., 2021), MoCoV2 (Chen et al., 2020b), and ConvNeXt (Liu et al., 2022) to fine-tune. See Appendix C and D for recipes.

## 4.2 IMAGENET EVALUATION

**Performance comparison with self-supervised transformers.** We first validate SparK on ImageNet with the pure convolutional model ConvNeXt (Liu et al., 2022). Smaller models {ViT, Swin, ConvNeXt}-S and the bigger ones {ViT, Swin, ConvNeXt}-B are compared separately. By comparing the results vertically in table 1, one can find the convolutional models, with SparK pre-training, overwhelmingly outperform transformer-based pre-training methods by large margins (+0.7~2.7), though SparK neither employs external models (DALL-E dVAE (Ramesh et al., 2021)), nor profits from advanced (MIM+CL) pre-training. This is somewhat surprising since transformers are well-known data-hungry models with much less inductive bias than convnets, and therefore are considered to benefit more from large-scale self-supervised training. The result here conveys a new message: convnets may have much more potential than expected, and their capability in visual representation may not be inferior to that of transformers. The key may depend on how to use powerful pre-training algorithms (*e.g.*, SparK or masked modeling) to turn this *potential* into *capability*.

**Efficiency.** Similar to MAE (He et al., 2021), SparK has the advantage of encoding efficiency, especially compared to contrastive learning that encodes two or more images in a forward pass. For instance, DINO and iBOT by default (Caron et al., 2021; Zhou et al., 2021) use multi-crop with 2 global crops of $224 \times 224$ and 10 locals of $96 \times 96$, leading to $2 + 10 (96/224)^2 \approx 3.8$ times the cost of single image encoding. In contrast, SparK requires only 40% of the theoretical overhead thanks to the sparsity of masked input: 60% of patches are masked, and sparse convolution only processes the rest. In practice, we found a sparse ResNet-50 can save ~23% memory footprint (26.4 GB *vs.* 34.5 GB for single batch size of 128). This allows us to train it on a 32GB Tesla V100, which otherwise is impossible for non-sparse pre-training. The efficiency also helps SparK scale up more easily.

Table 1: **Comparing SparK and self-supervised transformers on ImageNet.** All methods pre-train on ImageNet-1K an fine-tune with the resolution of 224. Top-1 validation accuracy is reported, the best results are in bold. "Extra model" indicates whether DALL-E's dVAE (trained on 250 million extra data) is used in pre-training. Entries with † are quoted from Zhou et al. (2021). ‡ is our reproduction using the official codes.

| Pre-training method | PT task | Enc. cost | Extra model | Small backbone Arch. | Acc. | Base backbone Arch. | Acc. |
|---|---|---|---|---|---|---|---|
| *Vision Transformer Backbone* | | | | | | | |
| MoCov3 (Chen et al., 2021) | CL | 5.0× | | ViT-S | 81.4 | ViT-B | 83.2 |
| DINO (Caron et al., 2021) | CL | 9.5× | | ViT-S | 82.0 | ViT-B | 82.8 |
| BEiT (Bao et al., 2021) | MIM | 2.5× | ✓ | ViT-S | 81.4† | ViT-B | 83.2 |
| CIM (Fang et al., 2022) | MIM | 2.5× | ✓ | ViT-S | 81.6 | ViT-B | 83.3 |
| CAE (Chen et al., 2022) | MIM | 2.5× | ✓ | ViT-S | 81.8 | ViT-B | 83.6 |
| MAE (He et al., 2021) | MIM | 0.6× | | ViT-S | 81.5‡ | ViT-B | 83.6 |
| SimMIM (Xie et al., 2021) | MIM | 2.5× | | ViT-S | 81.7 | ViT-B | 83.8 |
| iBOT (Zhou et al., 2021) | MIM+CL | 9.5× | | ViT-S | 82.3 | ViT-B | 84.0 |
| SimMIM (Xie et al., 2021) | MIM | 2.5× | | Swin-S | 83.4‡ | Swin-B | 84.0 |
| *Convolutional Backbone* | | | | | | | |
| SparK (ours) | MIM | 1× | | ConvX-S | **84.1** | ConvX-B | **84.8** |

---

[2]"Effective epoch" takes into account the total amount of images processed in pre-training. For instance, a typical contrastive learning encodes two images per forward pass, so the effective epoch is twice the literal value.

Table 2: **Comparing convnet-based SparK with transformer-based self-supervised learning on downstream tasks.** On ImageNet, the same fine-tuning resolution of 224 is used. On COCO, Mask R-CNN with FPN is equally applied. All methods follow a $3\times$ COCO schedule (36 epochs), while MAE fine-tunes longer (50 epochs). Average precisions of detection box ($AP^{bb}$) and segmentation mask ($AP^{mk}$) on `val2017` are reported. ‡ is reproduced using the official codes, since Liu et al. (2022) only runs models with Cascade Mask R-CNN.

| Pre-training method | Arch. | Eff.[2] epoch | Cls. Acc. | Det. $AP^{bb}$ | Det. $AP^{bb}_{75}$ | Seg. $AP^{mk}$ | Seg. $AP^{mk}_{75}$ |
|---|---|---|---|---|---|---|---|
| MoCov3 (Chen et al., 2021) | ViT-B | 1600 | 83.2 | 47.9 | – | 42.7 | – |
| BEiT (Bao et al., 2021) | ViT-B | 800 | 83.2 | 49.8 | – | 44.4 | – |
| Supervised (He et al., 2021) | ViT-B | 300 | 82.3 | 47.9 | – | 42.9 | – |
| MAE (He et al., 2021) | ViT-B | 1600 | 83.6 | 50.3 | – | 44.9 | – |
| *improvements over baseline* | | | **+1.3** | +2.4 | – | **+2.0** | – |
| Supervised (Liu et al., 2021) | Swin-B | 300 | 83.5 | 48.5 | 53.2 | 43.2 | 46.7 |
| SimMIM (Xie et al., 2021) | Swin-B | 800 | 84.0 | 50.4 | 55.5 | 44.4 | 47.9 |
| *improvements over baseline* | | | +0.5 | +1.9 | +2.3 | +1.2 | +1.2 |
| Supervised‡ (Liu et al., 2022) | ConvX-B | 300 | 83.8 | 47.7 | 52.6 | 43.2 | 46.6 |
| Spark (ours) | ConvX-B | 1600 | **84.8** | **51.2** | **56.1** | **45.1** | **48.9** |
| *improvements over baseline* | | | +1.0 | **+3.5** | **+3.5** | +1.9 | **+2.3** |

## 4.3 TRANSFERRING TO DOWNSTREAM TASKS

Previous results on ImageNet classification have exposed the potential of SparK pre-training. In this part we further evaluate the representation quality on fundamental downstream tasks, including object detection and instance segmentation on COCO (Lin et al., 2014). These tasks are challenging, serving as professional feature evaluators because they place higher demands than classification: models need to predict not only *what*, but also *where* the objects (instances) are. Here, we consider two different settings: comparison with self-supervised vision transformers, and then with convolutional networks. In all COCO experiments, we do not use advanced techniques such as multi-scale testing, large-scale jittering augmentation and soft-NMS. For more details on fine-tuning, see Appendix C and D.

**Performance *vs.* self-supervised transformers.** Table 2 compares the fine-tuning results on three downstream tasks: classification (Cls.), object detection (Det.), and instance segmentation (Seg.). Among all self-supervised methods, SparK is the best performer and the only one that pre-trains a convnet. Even when compared to the strongest SimMIM (Xie et al., 2021) with swin-transformer, SparK still yields superior results by $+0.8$, $+0.8$, $+0.7$ on three tasks respectively. It is particularly worth noting that without pre-training, ConvNeXt-B and Swin-B perform *similarly*. This indicates that the gains are *indeed* due to our SparK pre-training rather than the backbone difference.

Overall, it can also be seen that our approach exhibits the highest improvements over supervised baselines in table 2 (up to $+3.5\%$). All these observations are consistent with those in section 4.2 and once again validate that the BERT-style pre-training on convolutional networks is promising.

Table 3: **ResNet-50 results on downstream tasks.** SparK is compared to state-of-the-art contrastive learning algorithms. For ImageNet, the same training recipe from Wightman et al. (2021) (300-epoch fine-tuning with 224 resolution) is used. For COCO, Mask R-CNN ResNet50-FPN is equally fine-tuned for 12 or 24 epochs ($1\times$ or $2\times$), with average precision on `val2017` reported. SparK is highlighted as the only *generative* method.

| Pre-training (on ResNet-50) | Pre-train task | Eff. epoch | Cls. (Acc.) | $1\times$ Schedule $AP^{bb}$ | $1\times$ Schedule $AP^{mk}$ | $2\times$ Schedule $AP^{bb}$ | $2\times$ Schedule $AP^{mk}$ |
|---|---|---|---|---|---|---|---|
| Supervised | – | – | 79.8 | 38.9 | 35.4 | 41.3 | 37.3 |
| SimSiam (Chen & He, 2021) | Contrastive | 800 | 79.1 | – | – | – | – |
| MoCo (He et al., 2020) | Contrastive | 800 | – | 38.5 | 35.1 | 40.8 | 36.9 |
| MoCov2 (Chen et al., 2020b) | Contrastive | 1600 | 79.8 | 40.4 | 36.4 | 41.7 | 37.6 |
| SimCLR (Chen et al., 2020a) | Contrastive | 4000 | 80.0 | – | – | – | – |
| InfoMin (Tian et al., 2020) | Contrastive | 800 | – | 40.6 | 36.7 | 42.5 | 38.4 |
| BYOL (Grill et al., 2020) | Contrastive | 1600 | 80.0 | 40.4 | 37.2 | 42.3 | 38.3 |
| SwAV (Caron et al., 2020) | Contrastive | 1200 | 80.1 | – | – | 42.3 | 38.2 |
| SparK (ours) | Generative | 1600 | **80.6** | **41.6** | **37.7** | **43.4** | **39.4** |

**Performance *vs*. self-supervised convnets.** We then compare SparK to state-of-the-art convolutional contrastive learning methods. In table 3, all *contrastive* methods are basically on par with supervised pre-training. While SparK, the first *generative* pre-training method for hierarchical convnets, performs significantly better than them across all downstream tasks by +0.5~1.2 points. In particular, SparK does not rely on sophisticated augmentations which have proven to be essential for contrastive learning (Chen et al., 2020a; Tian et al., 2020). We attribute these superior results to the fact that generative pre-training (SparK) can inherently provide more supervisory signals than discriminative methods: it optimizes a reconstruction loss, a form of regression loss, which is considered to be more dense and localized than contrastive learning's instance classification loss.

**Feature transferability.** An intriguing phenomenon in table 2 and 3 is that the improvements over supervised baselines are more significant on COCO tasks than on ImageNet (+3.5 for ConvNet and +2.7 for ResNet). Notice there are several key differences between these two datasets: (i) the image resolution of COCO is much higher than that of ImageNet; (ii) most images in ImageNet are object-centric, while COCO images usually contain multiple disorganized objects. This *domain gap* poses a challenge for transfer learning, and SparK is demonstrated able to face it. This shows SparK can learn highly transferable features through the BERT-style generative pre-training.

## 4.4 Scaling up SparK

**We gradually scale up** the model size or training resolution and test SparK's performance. Results are reported in table 4, where we quote the accuracy of supervised baselines from Wightman et al. (2021) (the latest ResNet baselines) and Liu et al. (2022) (ConvNeXt) as "Baseline Acc.". As shown in the last column in table 4, one can observe that with our SparK pre-training, *all* models except ResNet-50 achieve performance on par with their non-pretrained versions of larger sizes. Such a qualitative leap indicates SparK can push a convnet to the "next level" in terms of representation capability. Comparing the results horizontally, SparK improves all supervised baselines by large margins of +0.8~1.7, verifying such a *self-supervised* learning can make better use of model capacity than *supervised* pre-training in this evaluation. Overall, the results demonstrate a favorable scaling ability of SparK as larger models benefit more. The steady gains across classical and modern architectures also make us believe SparK can boost many other state-of-the-art convolutional networks like VAN (Guo et al., 2022), RepLKNet (Ding et al., 2022), and InternImage (Wang et al., 2022).

Table 4: **Scaling up SparK with model size and training resolution.** ImageNet top-1 accuracy is reported. Absolute improvements over baselines are listed as $\Delta$. The last column indicates whether *SparK's* performance with a smaller model (*e.g.*, 84.1 of ConvNeXt-S) reaches the *baseline* of a larger one (*e.g.*, 83.8 of ConvNeXt-B).

| Architecture | Reso. | #Para. (M) | FLOPs (G) | Baseline Acc. | Spark Acc. | $\Delta$ | Reach the next level |
|---|---|---|---|---|---|---|---|
| *Classical Architecture* | | | | | | | |
| ResNet-50 | 224 | 25.6 | 4.1 | 79.8 | 80.6 | +0.8 | ✗ |
| ResNet-101 | 224 | 44.5 | 7.9 | 81.3 | 82.2 | +0.9 | ✓ |
| ResNet-152 | 224 | 60.2 | 11.6 | 81.8 | 82.7 | +0.9 | ✓ |
| ResNet-200 | 224 | 64.7 | 15.1 | 82.1 | 83.1 | +1.0 | − |
| *Modern Architecture* | | | | | | | |
| ConvNeXt-Small | 224 | 50.0 | 8.7 | 83.1 | 84.1 | +1.0 | ✓ |
| ConvNeXt-Base | 224 | 89.0 | 15.4 | 83.8 | 84.8 | +1.0 | ✓ |
| ConvNeXt-Large | 224 | 198 | 34.4 | 84.3 | 85.4 | +1.1 | − |
| ConvNeXt-Large | 384 | 198 | 101 | 84.3 | **86.0** | **+1.7** | − |

## 4.5 Ablation Study

In this study, we gradually ablate the components in SparK framework and check the corresponding performance respectively. ImageNet fine-tuning results of each SparK's variants are listed in table 5.

**Core designs.** We first remove the two most important designs in SparK: sparse masking strategy and hierarchical architecture. By replacing our sparse strategy with the zero-outing discussed in section 3.1, we observe a noticeable performance degradation in row 3 of table 5 that almost *reaches* the supervised baseline. This suggests the issues raised by zero-outing (like data distribution shift in figure 1 and mask pattern vanish in figure 3) can lead to ineffective pre-training. We then remove the hierarchical design (row 4), which results in a *single-scale* masked modeling that is commonly used

Table 5: **The ablation study on the importance of each components in SparK**. Experiments are based on ConvNeXt-Small, with ImageNet validation accuracy reported. Our default setting is in row 2. Differences are highlighted in blue. "APE": absolute positional embedding; "std.": standard deviation of four experiments.

| | Method | Masking | Hierarchy | APE | Loss | Epoch | Acc. | Δ | std. |
|---|---|---|---|---|---|---|---|---|---|
| 1 | Not pretrained | | | | | | 83.1 | -1.0 | |
| 2 | SparK (ours) | sparse | ✓ | ✗ | masked only | 1600 | 84.1 | 0.0 | 0.07 |
| 3 | zero-outing | zero-outing | ✓ | ✗ | masked only | 1600 | 83.2 | -0.9 | 0.06 |
| 4 | w/o hierarchy | sparse | ✗ | ✗ | masked only | 1600 | 83.6 | -0.5 | 0.04 |
| 5 | w/ APE | sparse | ✓ | ✓ | masked only | 1600 | 83.9 | -0.2 | 0.10 |
| 6 | w/ more loss | sparse | ✓ | ✗ | all | 1600 | 83.3 | -0.8 | 0.12 |
| 7 | pre-train less | sparse | ✓ | ✗ | masked only | 800 | 83.7 | -0.4 | 0.05 |

for transformers (Devlin et al., 2018; Bao et al., 2021; He et al., 2021; Xie et al., 2021). It only uses the features at the end of encoder to reconstruct. This modification is shown to impair the fine-tuning performance as well. In sum, both sparse strategy and hierarchy design play key roles in SparK.

**Other components.** In addition, we find adding absolute positional embeddings (row 5) is practically useless for learning convolutional representations. We also observe calculating loss values only on masked patches gives higher accuracy (row 6), which is consistent with He et al. (2021). Finally and reasonably, our SparK benefits from longer pre-training as verified in row 7.

## 4.6 VISUALIZATION

We visualize some reconstruction results to check how the model performs in pre-training. From figure 4 we can see that the model is able to make different but plausible predictions on masked regions (*e.g.*, in the 2-nd column). In the 4-th and 6-th columns, the model can almost reconstruct the round shape of red fruits from the very small portion of exposed edges. The clear texture in the 3-rd column also shows the model can capture the visual signals with medium or high frequencies.

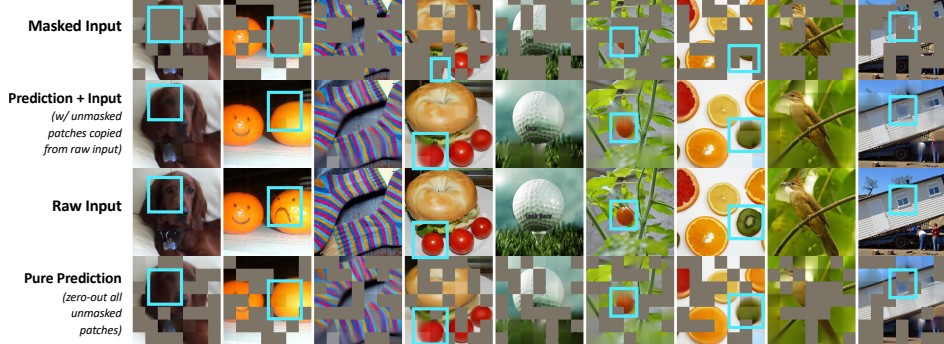

Figure 4: **Reconstruction examples by a pre-trained ConvNeXt-Base with a mask ratio of** 60%. Images are randomly selected from ImageNet validation set. Several interesting regions are highlighted.

## 5 CONCLUSION

The field of natural language processing (NLP) has witnessed the rise and proliferation of masked modeling in *NLP transformers*. More recently, there have been efforts to extend this paradigm to *vision transformers*, although its application to *convnets* has proven problematic. This has spurred us to investigate the fundamental differences between language and image processing, and motivated us to propose a solution: SparK. SparK involves treating unmasked patches as sparse voxels, and encoding them using sparse convolution. Additionally, we employ a hierarchical decoder to fully leverage the benefits of convnet's hierarchy. SparK enables masked modeling to be applied effectively to any convnet, and results in a substantial performance increase on downstream tasks. Our research showcases the potential of BERT-style pre-training on convnets and is an initial step towards its future implementation. We hope our findings will inspire further exploration of generative pre-training on convnets, and facilitate the adoption of the pretrain-finetune paradigm in computer vision community.

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

## A  Details: Decoder Architecture

SparK is a general method that does not limit the specific encoder to be pre-trained. In other words, the definition of the encoder is all up to the user (*e.g.*, a standard ResNet-50). In the implementation presented in section 4.1, the only undefined component is the decoder. We thus give its PyTorch implementation as follows. In our experiments, the same decoder of `LightDecoder(768, 32)` is used equally for all encoders, including different ResNets and ConvNeXts.

```python
import math
import torch.nn as nn

class UNetBlock2x(nn.Module):
    def __init__(self, cin, cout):
        super().__init__()
        self.b = nn.Sequential(
            nn.Conv2d(cin, cin, kernel_size=3, stride=1, padding=1, bias=False),
            nn.BatchNorm2d(cin), nn.ReLU6(inplace=True),
            nn.Conv2d(cin, cout, kernel_size=3, stride=1, padding=1, bias=False),
            nn.BatchNorm2d(cout),
        )

    def forward(self, x):
        return self.b(x)

class DecoderConv(nn.Module):
    def __init__(self, cin, cout):
        super().__init__()
        self.up = nn.ConvTranspose2d(cin, cin, kernel_size=4, stride=2, padding=1,
            bias=True)

        self.conv = UNetBlock2x(cin, cout)

    def forward(self, x):
        x = self.up(x)
        return self.conv(x)

class LightDecoder(nn.Module):
    def __init__(self, decoder_fea_dim, upsample_ratio):
        super().__init__()
        self.fea_dim = decoder_fea_dim

        n = round(math.log2(upsample_ratio))
        channels = [self.fea_dim // 2 ** i for i in range(n + 1)]
        self.dec = nn.ModuleList([
            DecoderConv(cin, cout) for (cin, cout) in zip(channels[:-1],
                channels[1:])
        ])
        self.proj = nn.Conv2d(channels[-1], 3, kernel_size=1, stride=1, bias=True)

    def forward(self, to_dec):
        x = 0
        for i, d in enumerate(self.dec):
            if i < len(to_dec) and to_dec[i] is not None:
                x = x + to_dec[i]
            x = self.dec[i](x)
        return self.proj(x)
```

## B    ADDITIONAL RESULTS: LINEAR EVALUATION

We report the small-sized models' linear evaluation performance in table 6. In this evaluation protocol, the pre-trained backbone model is frozen and only a linear projection head would be fine-tuned. This protocol is all the rage in contrastive learning (Chen et al., 2020a; He et al., 2020; Chen & He, 2021; Caron et al., 2021), which can probe the linear separability of deep representations, and has been quite popular in computer vision due to the richness of image data augmentations compared to other modalities (Cubuk et al., 2019; Tian et al., 2021; Cheng et al., 2022). Note that MoCoV3 (Chen et al., 2021) is the only contrastive learning method in table 6, which aims to learn a global representation, and is therefore more suitable than non-contrastive methods on tasks like linear evaluation. SparK shows its decent performance compared to other non-contrastive methods.

Table 6: **Linear evaluation results.** Numbers of other work are directly quoted form Chen et al. (2022).

| Method | BEiT | CAE | SparK | MoCoV3 |
|---|---|---|---|---|
| Contrastive | ✗ | ✗ | ✗ | ✓ |
| Accuracy (%) | 15.7 | 51.8 | 54.7 | 73.1 |

## C    DETAILS: IMAGENET FINE-TUNING

We refer to the latest open-source ResNet baseline of Wightman et al. (2021) to fine-tune ResNets. For ConvNeXts Liu et al. (2022), we simply use their official implementation. Since the original configurations in Wightman et al. (2021); Liu et al. (2022) are based on supervised training from scrach, we adjust some hyperparameters for doing fine-tuning. Details are given in table 7 and table 8.

Table 7: **ImageNet fine-tuning recipe for ResNets, referring to Wightman et al. (2021).**

| Configuration | Value | Configuration | Value |
|---|---|---|---|
| Image resolution | 224 | Epochs | 300 |
| Test image crop | 0.95 | Batch size | 2048 |
| Optimizer | LAMB | Learning rate | 8e-3 |
| Scheduler | Consine | Weight decay | 0.02 |
| Repeated aug. | ✓ | Dropout | ✗ |
| Rand aug. | 7/0.5 | Stoch. depth | ✓ |
| Gradient clip. | ✗ | BCE loss | ✓ |
| Mixup alpha | 0.1 | Label smoothing | 0.1 |
| Cutmix alpha | 1.0 | EMA | {0.99,0.999} |

Table 8: **ImageNet fine-tuning recipe for ConvNeXts, referring to Liu et al. (2022).**

| Configuration | Value | Configuration | Value |
|---|---|---|---|
| Image resolution | 224 | Epochs | 200 |
| Test image crop | 0.95 | Batch size | 2048 |
| Optimizer | AdamW | Learning rate | 3.2e-3 |
| Scheduler | Consine | Weight decay | 0.01 |
| Repeated aug. | ✓ | Dropout | ✗ |
| Rand aug. | 9/0.5/inc1 | Stoch. depth | ✓ |
| Gradient clip. | ✗ | BCE loss | ✗ |
| Mixup alpha | 0.8 | Label smoothing | 0.1 |
| Cutmix alpha | 1.0 | EMA | {0.99,0.999} |

## D  DETAILS: COCO FINE-TUNING

On COCO, we use the official implementations of MoCoV2 (Chen et al., 2020b) and ConvNeXt (Liu et al., 2022) to evaluate ResNets and ConvNeXts. These implementations are based on Detectron2 (Wu et al., 2019) and MMDetection (Chen et al., 2019) respectively. Following the convention, we do not use advanced techniques like multi-scale testing, large-scale jittering augmentation, or soft-NMS, in all our COCO experiments for fairness. Details are in table 9 and table 10.

Table 9: **COCO fine-tuning configuration for ResNets, referring to the standard implementation of MoCoV2 (Chen et al., 2020b).** Mask R-CNN with FPN is used. $x = A$ for the so-called "A×" schedule. For instance, a $2\times$ fine-tuning schedule means a 24-epoch training with 0.1-epoch warm-up.

| Configuration | Value | Configuration | Value |
|---|---|---|---|
| Image resize | (384, 600) | Normalization mean | [123.7, 116.3, 103.5] |
| Multi-scale testing | ✗ | Normalization std | [58.4, 57.1, 57.4] |
| Large-scale jittering aug | ✗ | Optimizer | AdamW |
| Soft-NMS | ✗ | Weight decay | 0.0001 |
| Epochs | $12x$ | Learning rate (LR) | 2e-4 |
| Warm-up epochs | $0.05x$ | LR layer decay | 0.65 |
| LR scheduled epochs | $[9x, 11x]$ | LR scheduled ratio | 0.2 |

Table 10: **COCO configuration for ConvNeXts, referring to the standard implementation of ConvNeXt (Liu et al., 2022).** Following the convention of self-supervised learning, $3\times$ Mask R-CNN with FPN is used.

| Configuration | Value | Configuration | Value |
|---|---|---|---|
| Image resize | (1333, 800) | Normalization mean | [123.7, 116.3, 103.5] |
| Multi-scale testing | ✗ | Normalization std | [58.4, 57.1, 57.4] |
| Large-scale jittering aug | ✗ | Optimizer | AdamW |
| Soft-NMS | ✗ | Weight decay | 0.05 |
| Epochs | 36 | Learning rate (LR) | 2e-4 |
| Warm-up epochs | 0.15 | LR layer decay | 0.65 |
| LR scheduled epochs | [27, 33] | LR scheduled ratio | 0.2 |

