# OpenReview forum: "Designing BERT for Convolutional Networks: Sparse and Hierarchical Masked Modeling"
_ICLR.cc/2023/Conference — ICLR 2023 notable top 25%_

### Official Review · Reviewer_71p6 · 2022-10-24

**Confidence:** 4
**Correctness:** 3
**Technical Novelty And Significance:** 3
**Empirical Novelty And Significance:** 3
**Recommendation:** 6

**Clarity, Quality, Novelty And Reproducibility:**

- **Clarity:** The paper overall is easy to read and follow.

- **Quality:** Good.

- **Novelty:** Although the idea of using sparseconv to handle irregular input is not new for both 2D and 3D tasks, it seems new to use it for improving MIM of convnets.

- **Reproducibility:** Code is not attached in the submission. Many important details are missing. It would be difficult to reproduce the results based on the information provided in the current submission.


**Strength And Weaknesses:**

Strengths:

- Transferring the successful experiences of MIM from Transformers to convolutional models is an important and potentially impactful topic. This paper exhibits some promising results of applying MIM to ConvNeXt and ResNet.

- The idea of using the submanifold sparse convolution to handle irregular masked input is well-motivated and new. The hierarchical encoding and decoding design also look interesting.

Weaknesses:

- The motivation shown in Figure 1 may be straightforward but a bit misleading. I think the main advantage of the proposed method based on the submanifold sparse convolution is that the mask patterns are kept in all intermediate features. If we use the "vanilla sparseconv" mentioned in Sec 2.3, the model will also be difficult to generalize to raw input even if the input data distributions are close.

- There are several missing references that should be discussed. [r1] is a widely used sparseconv framework in 3D understanding. [r2] is an existing method that uses sparseconv to accelerate 2D image recognition.

[r1] 4D Spatio-Temporal ConvNets: Minkowski Convolutional Neural Networks, CVPR 2019

[r2] Dynamic convolutions: Exploiting spatial sparsity for faster inference, CVPR 2020.

- Many important implementation details are missing. Sec. 4.1 introduces the details of the proposed pre-training method. However, there is no detail about the fine-tuning process for ImageNet and downstream tasks, which is very important information to compare different pre-training methods and reproduce the results. For example, MAE and SimMIM report the results with 50/100-epoch finetuning. Can SparK models achieve competitive results on ImageNet with fewer finetuning epochs than standard training (300 epochs)? Are extra tricks used during fine-tuning?

- The discussions in Sec 4.2 emphasize the encoding efficiency of the proposed method. From my experience, sparseconv is usually much slower than standard convolutions with similar theoretical complexity since the standard convolution is well-supported and highly optimized on GPUs. I guess 60% masked patches may not lead to significantly higher training speed than models without masked input in practice. The encoding cost shown in Table 1 is very ambiguous. I think it would be better to report the wall-clock time of the whole system and training process like the MAE paper to clearly compare different methods.

- The comparisons of small backbones are unfair.  ConvNeXt-S have doubled computational costs/parameters compared to ViT-S/Swin-T.

- Why are the detection frameworks used for small and base backbones in Table 2 different? Besides, it would be better to report the performance of the baseline supervised ConvNeXt models in both settings.

- According to Table 4, it seems ConvNeXt-L cannot outperform MAE (ViT-L). It would be better to add more discussions and analyses of this.

Minor issues:

- How about the results if we zero out features of the masked regions (before all 7x7 convs) or replace them with the mask embeddings instead of zeroing out input? Considering sparseconv requires extra libraries/extensions and might not be easy to use, this implementation can be a competitive baseline.

- The formats of citations are different in some sentences. For example, in Section 2.2, there are both "Bao et al. (2021)" and "(Devlin et al., 2018)".

**Summary Of The Paper:**

This paper presents a masked modeling method for convolutional vision backbones. Instead of simply zeroing out the masked regions in the input images, the method exploits the submanifold sparse convolution to handle irregular masked input. The method is evaluated on multiple popular vision tasks and backbone models.

**Summary Of The Review:**

The method presented in this paper is interesting and may be useful in future research and applications. However, there are still many important issues that require further clarification and discussion including missing comparisons, baselines and experimental details. I would upgrade my rating if the above-mentioned issues are addressed.

---

> ### Author Response · Authors · 2022-11-09
> **Official Response to Reviewer 71p6 [2/2]**
>
> (here is the second part of our response)
>
> > [Minor 1] More ablations.
>
> A: We actually adopted the suggestion of using learnable mask tokens as a baseline (using a 768-dim vector to replace a masked patch of 3x32x32=768 pixels). However, we found the pre-training is unstable, with "dip" caused in the curve of reconstruction loss. This phenomenon indicates an optimization problem for the deep model under pre-training. We believe this may be due to the fact that convnet is not adapted to the "tokenized" input, or it may also be linked to the optimizer used. We leave this issue for the future.
>
> > [Minor 2] Citations are different in some sentences.
>
> A: Thanks for the suggestion. We have fixed the citations and checked the full text.

---

> > ### Comment · Reviewer_71p6 · 2022-11-22
> > **Thanks for your response**
> >
> > I would like to thank the authors for the detailed feedback and additional results. The response addressed my concerns about the comparisons and the fine-tuning details. But there are still some unresolved issues about the efficiency of the training process and the scaling ability of the method:
> > - The sparse convolution seems cannot largely speed up the training process like MAE for ViT and the model requires 300 epoch fine-tuning that is the same as the configuration of training from scratch.
> > - It is still unknown whether the method can enable training very large ConvNeXt on ImageNet.
> >
> > Overall, although there are still several unresolved issues, I think this paper provides a pioneering solution and some useful results on pre-training CNNs with MIM, which may inspire future research and applications. Therefore, I would like to upgrade my rating to 6.

---

> > > ### Author Response · Authors · 2022-12-13
> > > **Thank you! More response and updates.**
> > >
> > > **To the three more concerns:**
> > >
> > > > [U1] "the sparse convolution seems cannot largely speed up the training process like MAE for ViT"
> > >
> > > A:  Although transformers can be inherently better at processing variable-length (sparse) data than conv, here we still provide an idea trying to reduce this gap: given the sparse pattern in SparK is more structural than that in point cloud data (e.g., it is block-wise and has a constant sparse ratio), we believe faster sparseconv can be designed for SparK (e.g., by preprocessing the Rulebook in $\texttt{im2col}$).
> > >
> > > > [U2] "classification finetuning epochsare the same as from scratch".
> > >
> > > A: We would like to highlight the detection and segmentation tasks where we **compared SparK with supervised pretraining instead of baselines from scratch**. Also, these tasks are considered to be closer to real downstream than classification.
> > > Our improvements on them are more substantial (up to 3.6AP) than classification, which can better justify our finetuning effectiveness.
> > >
> > > > [U3] "It is still unknown whether the method can enable training very large ConvNeXt on ImageNet"
> > >
> > > A: We have made every effort in this rebuttal period to pretrain with heavier settings. Since convnext-XL is too expensive for us, we try to train convnext-L in higher resolution (224->384) and found:
> > >
> > > |convnext|resolution|accuracy|
> > > |:-:|:-:|:-:|
> > > |large|224|85.4|
> > > |large|384|86.0|
> > >
> > > One can see the performance is still scaled with higher computational budget on large models. Taken this trend and our scalability Table 4 (larger convnext gains more), we believe it is promising to scale this *pioneering solution* to convnext-XL or even more heaviers. We will do our best to study this in the future.
> > >
> > > &nbsp;
> > >
> > > ---------
> > >
> > > **Last**, thanks so much for helping us improve this work through your constructive and professional perspective, and appreciate your comment of "pioneering solution"!
> > > Hope our further response can address your concerns above :). Although the author response period is coming to an end, we will continue to keep the discussion if we can.

---

> > > > ### Public Comment · ~Dejan_Štepec1 · 2023-08-10
> > > > **Finetuning epochs**
> > > >
> > > > Dear authors,
> > > >
> > > > with regards to U2 from the reviewer 71p6: "The sparse convolution seems cannot largely speed up the training process like MAE for ViT and the model requires 300 epoch fine-tuning that is the same as the configuration of training from scratch."
> > > >
> > > > I think that the reviewer had in mind that for MAE, the authors only fine-tuned for 100 (ViT-B) and 50 (ViT-L/H) epochs (MAE paper, A.1, Table 9). The supervised models that were trained from scratch in the MAE paper were trained for 300 epochs (MAE paper, A.2, Table 11).
> > > >
> > > > SparK was fine-tuned for 300 epochs, which is the same configuration as training supervised ViTs from scratch. MAE thus achieved comparable performance with 3-6x less fine-tuning epochs.
> > > >
> > > > Can you report your results also with the same fine-tuning configuration (50/100 epochs) as used in MAE? Only then the comparison can be fair.

---

> > > > > ### Author Response · Authors · 2023-10-02
> > > > > **Downstream evaluation**
> > > > >
> > > > > Thank you for your advice and interest in our work.
> > > > >
> > > > > Objectively speaking, the best finetuning recipes or hyperparameters for ConvNets and ViTs are *different*. Our recipes are borrowed from AAMIM [1] rather than MAE, because the former also pretrains ConvNets while the latter is only for ViTs. And in MAE they also did a *grid search* on finetuning hyperparameters. So using ViT's best recipe to finetune a Convnet (or vice versa) may not lead to convincing conclusions.
> > > > >
> > > > > On the other hand, we would suggest *focusing more* on the performance improvements on *REAL* downstream tasks like COCO object detection. Paying too much attention on ImageNet pretraining-and-finetuning can lead us into a situation similar to "overfitting". On COCO, SparK shows much better performance than MAE (MIM on ViT) and SimMIM (MIM on Swin-Transformer), which is a solid proof of SparK's effectiveness in "making MIM also successful on ConvNets".
> > > > >
> > > > >
> > > > > ---------------------
> > > > > [1] Li, Siyuan, et al. "Architecture-Agnostic Masked Image Modeling--From ViT back to CNN." arXiv preprint arXiv:2205.13943 (2022).

---

> ### Author Response · Authors · 2022-11-09
> **Official Response to Reviewer 71p6 [1/2]**
>
> Many thanks to Reviewer 71p6 for their professional, detailed, and valuable reviews. We note that all the weaknesses, while not devaluing **our core contribution of "extending the BERT-style pre-training to convnets"**, are very professional and would contribute a lot to the refinement of our manuscript. Below we have listed our responses to each of the comments:
>
> ------------
>
> > [Weakness 1] The motivation shown in Figure 1 may be straightforward but a bit misleading.
>
> A: We highly agree with the benefit of using *"Submanifold"* Sparse Convolution (SSC) for BERT pre-training on convnets, which is one of SparK's key contributions to the field.  We have revised Introduction/Figure 1 in manuscript that refers the readers to Figure 3 and Section 3.1 to highlight the advantage of submanifold sparse convolution. We believe this can better illustrate our motivation and contribution about using SSC.
>
> > [W2] Missing references that should be discussed.
>
> A: Thanks for the suggestion. These works are indeed highly related to our use of sparse convolution for 2D masked image modeling. We have cited them and updated the Related Work in manuscript accordingly.
>
> > [W3] No details on fine-tuning.
>
> A: We have mentioned that we follow the standard fine-tune recipes of RSB-A2 for ImageNet fine-tuning, and 3x Cascade/Mask-RCNN for COCO fine-tuning, which are taken from [r1] and MMDetection [r2] respectively. We did not report these recipes in origin manuscript due to the limited space. For convenience, we have listed them now in the Appendix, which can be checked.
>
> > [W4] No wall-clock efficiency demonstration.
>
> A: Time or memory efficiency matters a lot for better scaling pre-training algorithms. We agree the very professional comment that sparse convolution is usually slower than ordinary convolution that is highly optimized on specific hardwares. We also observed that the efficiency improvement of SparK is mainly in **memory** (e.g., it allows us to pre-train with larger batch sizes on 32GB Tesla V100). This memory efficiency is consistent with common observations in the field of 3D vision [r3, r4, r5]. We have post our observations in Section 4.2.
>
> > [W5] ConvNeXt-S have doubled computational costs/parameters compared to ViT-S/Swin-T.
>
> A: We have updated the table in revision (using Swin-S vs. ConvNeXt-S) for a more fair comparison. Furthermore, we also note that SparK-based ConvNeXt-**S** outperforms all ViT-**B**/Swin-**B** in Table 1. Taken these facts, the validity of SparK is still well justified.
>
> > [W6] (i) Different settings of small/base backbons in Table 2. (ii) Better to report baseline of ConvNeXt-B in Table 2.
>
> A: (i) For reliability and fairness, we have done our best to gather as many, commonly-used fine-tuning configurations as possible (from BEiT, iBOT, SimMIM, MAE, etc.), and found the most commons are 3x Cascade RCNN for small-sized models and 3x Mask-RCNN for the base-sized. (ii) We have added the ConvNeXt-B baselines in Table 2 and also include them here, which shows a leap in performance by SparK pre-training:
>
> |Method|Cls. (\%)|Det.(AP$^\text{bb}$)|Det.(AP$^\text{bb}_{75}$)|Seg.(AP$^\text{bb}$)|Seg.(AP$^\text{bb}_{75}$)|
> |:-:|:-:|:-:|:-:|:-:|:-:|
> |Supervised|83.8| 47.7| 52.6| 43.2| 46.6|
> |SparK|84.8| 51.2| 56.1| 45.1| 48.9|
>
> --------------
>
> > [W7] In Table 4, it seems ConvNeXt-L cannot outperform MAE (ViT-L).
>
> A: It is noted that MAE optimizes the pre-training hyperparameters heavily by grid search, e.g., in Figure 5 and Table 1 (a,b,c,d,e,f) of [r6]. Such exploration is useful for the domain, but we were limited by the computational resource and thus used **the same** pre-training configurations on all ResNets and ConvNeXts. It can be believed that a simple work with clear room for optimization is also of great benefit to our field. And we will also do our best to optimize, which will be one of the most primary goals of our future work.
>
> &nbsp;
>
> ------------------------------
>
> [r1] Wightman, Ross, Hugo Touvron, and Hervé Jégou. "Resnet strikes back: An improved training procedure in timm." arXiv preprint arXiv:2110.00476.
>
> [r2] Chen, Kai, et al. "MMDetection: Open mmlab detection toolbox and benchmark." arXiv preprint arXiv:1906.07155 (2019).
>
> [r3] Rocco, Ignacio, Relja Arandjelović, and Josef Sivic. "Efficient neighbourhood consensus networks via submanifold sparse convolutions." European conference on computer vision. Springer, Cham, 2020.
>
> [r4] Zhou, Yin, and Oncel Tuzel. "Voxelnet: End-to-end learning for point cloud based 3d object detection." Proceedings of the IEEE conference on computer vision and pattern recognition. 2018.
>
> [r5] Srinivas, Suraj, Akshayvarun Subramanya, and R. Venkatesh Babu. "Training sparse neural networks." Proceedings of the IEEE conference on computer vision and pattern recognition workshops. 2017.
>
> [r6] He, Kaiming, et al. "Masked autoencoders are scalable vision learners." Proceedings of the IEEE/CVF Conference on Computer Vision and Pattern Recognition. 2022.

---

### Official Review · Reviewer_tYbJ · 2022-10-24

**Confidence:** 4
**Correctness:** 4
**Technical Novelty And Significance:** 3
**Empirical Novelty And Significance:** 3
**Recommendation:** 8

**Clarity, Quality, Novelty And Reproducibility:**

The approach is simple, the authors describe it in detail and provide code of the decoder. It should not be difficult to reproduce the results.
Questions to the authors:
- How well is SparK able to reconstruct images compared to Transformer? Do they reach similar MSE test error?
- How well does the method work with linear probing? An evaluation in linear probing mode is missing, even though it is common for contrastive methods.

**Strength And Weaknesses:**

Strengths:
- well written and easy to follow
- strong results on using networks pretrained with the proposed method for object detection, which could be attributed to the tested object detection architecture, Mask R-CNN, designed for convolutional networks.
- detailed ablation study. Would be interesting to see if object detection sees improvements when using zero-out masking mode.
- the provided PyTorch code is useful to have better understanding of the approach

Weaknesses
- the authors do not show results on large backbones with 86+ classification accuracy. It is not clear if the approach does not scale to larger backbones, or the authors were limited by computational resources. Including such results would be very useful for future work.
- no statistical significance evaluation, which makes validating the claims difficult, especially in the ablation study. What is the variance in the experiment?

**Summary Of The Paper:**

The authors propose a method of pre-training convolutional neural networks via masking. They mask parts of the input image, take a convolutional network as encoder, and add a U-net style decoder, with which they predict the parts which were masked out. After pre-training, the decoder is removed, and the encoder is fine-tuned for downstream tasks, similar to recent Transformer-based approaches.

The approach is experimentally validated on ImageNet classification and COCO object detection tasks and shows improvements over other contrastive and masked pre-training methods with models of similar size in terms of the number of parameters and computational complexity.

**Summary Of The Review:**

I suggest accept, seems like a good paper. It would be beneficial to have linear probing results in addition to fine-tuning.

---

> ### Author Response · Authors · 2022-11-09
> **Official Response to Reviewer tYbJ**
>
> Thank you for the helpful comments and suggestions. See below for the answers to your questions and comments.
>
> > [Weakness 1] No large backbones with 86+ accuracy.
>
> A: Before replying we would like to highlight our core contribution is that, to our knowledge, **this work is the first successful application of BERT (masked modeling) to convolutional networks**. In a sense, SparK plays a role of "explorer", so as you say it still has the room for improvement :-).
> - We highly agree that a well-optimized pre-trained large model can be pretty useful to our community.
> - We would also like to highlight a clear trend that SparK **brings more improvements for larger models**, which shows the scalability of SparK as listed in Table 4.
> - Due to the limitation of computation resources, we excluded the experiments that: (i) involve hyperparameter tuning of pre-training (we used **the same settings** on all ResNets and ConvNeXts), and (ii)  with the image resolution larger than $224$.
> - Thus, we believe SparK has clear room for optimization. E.g., we can optimize SparK's hyperparameters, or using the $384$ resolution (for instance, the BEiT-large [r1] got an accuracy of 86.3 under a $384$ setting). We will do our best to optimize, and this will be one of the primary goals of our future work.
>
> > [Weakness 2] No statistical significance evaluation.
>
> A: Thanks for this professional advice. We have now reported the experimental variance in Section 4.5. For SparK on ConvNeXt-S, the standard deviation of the ImageNet val-accuracy of four experiments is $0.07$.
>
> > [Clarity 1] Report MSE error could be useful.
>
> A: We report our specific MSE errors below and compare them to the MAE [r2]. Note that the specific values may only be of some reference significance because MAE and SparK use different patch sizes.
>
> |Method|Patch Size|Recon. Loss|
> |:-:|:-:|:-:|
> |MAE|$16\times16$|0.4234|
> |SparK|$32\times32$|0.4506|
>
> > [Clarity 2] Report linear evaluation performance could be useful.
>
> A: We report the small-sized models' linear evaluation performance below. Numbers are directly quoted from [r3]. MoCoV3 is the only **contrastive** learning method in comparison, which inherently learns a global representation and is therefore more suitable than **non-contrastive** methods for tasks such as linear probing. SparK shows its good performance relative to other non-contrastive ones. We have also reported this resutls in Appendix.
>
> |||||||
> |:-|:-|:-|:-|:-|:-|
> |**Method**|BEiT|CAE|SparK|MoCoV3|
> |**Accuracy**|15.7|51.8|54.7|73.1|
>
>
> &nbsp;
>
> -----------------------
> [r1] Bao, Hangbo, et al. "BEiT: BERT Pre-Training of Image Transformers." International Conference on Learning Representations. 2021.
>
> [r2] He, Kaiming, et al. "Masked autoencoders are scalable vision learners." Proceedings of the IEEE/CVF Conference on Computer Vision and Pattern Recognition. 2022.
>
> [r3] Chen, Xiaokang, et al. "Context autoencoder for self-supervised representation learning." arXiv preprint arXiv:2202.03026 (2022).

---

> > ### Author Response · Authors · 2022-12-13
> > **Thank you! We have more updates and expect more discussions**
> >
> > We would like to express our sincere appreciation for your valuable comments, which have greatly complemented and improved our paper.
> > No we have more results on your **[Weakness 1]** "No large backbones with 86+ accuracy."
> >
> > We have done our best in the rebuttal period to pretrain with heavier settings. Since convnext-XL is too expensive for us, we try to train convnext-L in higher resolution (224->384). The results are as follows:
> >
> > |convnext|resolution|accuracy|
> > |:-:|:-:|:-:|
> > |large|224|85.4|
> > |large|384|86.0|
> >
> > The performance is still scaled with higher computational budget on large models. Taken this trend and our scalability Table 4 (larger convnext gains more), we believe it is possible to continue scaling SparK to 86+ accuracy. We will make every effort to study this in the future.
> >
> > &nbsp;
> >
> > ---------
> >
> > Again, thank you for the helpful comments and suggestions. Hope our further response can address your concerns above :). Although the author response period is coming to an end, we will continue to keep the discussion if we can.

---

### Official Review · Reviewer_Zhkz · 2022-10-25

**Confidence:** 5
**Correctness:** 3
**Technical Novelty And Significance:** 2
**Empirical Novelty And Significance:** 2
**Recommendation:** 5

**Clarity, Quality, Novelty And Reproducibility:**

Clarity, the paper is well-organized and easy to read.


Novelty: The originality and novelty of this paper are very limited. This paper follows the way of MAE, SimMIM, and BEiT.  Compared to BEiT, MAE, and SimMIM prove that simply reconstructing the mask region (l_1 and l_2 norm) will be good enough for the pre-training. This paper transfers the same idea to the ConvNets by incorporating sparse convolution. Therefore, the contribution to the community is much less than the published ones.

Reproducibility: It could be able to reproduce the paper, however, some details are not clear.

**Strength And Weaknesses:**

Pro:
1. This paper solves the patch mask on the ConvNet by using a "sparse" convolution operation. At the same time, the authors also incorporate the U-Net-like design, which passes the encoding mask to the decoding part to preserve the mask information.

2. This paper provides strong performance across all the tasks and architecture in a self-supervised learning setting.


Cons:

1. The authors seem to overclaim several two important contributions.

a. Removing information of masked parts is difficult for ConvNet. This has been identified by submanifold sparse convolution, which aims to preserve the structure of the unmasked regions. Therefore, it should not be a difficulty anymore.

b. The author claims in the experiment that their work proves that the convnet is not inferior to the transformers. This seems wrong to me. I do think ConvNext has already shown that.

2. In Fig.1, since the second row is the histogram, I can understand that using zeros as masks could increase the number of zeros, but I do not understand how the sparsely dropping does not decrease the frequency of other pixels (Fig. 1c).

3. Table 5 shows that even without the hierarchical encoding, ConvNext outperforms the transform architectures. Indeed, the hierarchical encoding works, but not impressive. The performance gained mostly due to ConvNext itself.





**Summary Of The Paper:**

This paper works on the pretext task learning a subfield of self-supervised learning. It follows the track of predicting the original image given random non-overlapped masks. To make it work on non-transformer architectures, the authors propose to use a hierarchical and sparse convolution network to pre-train the network with the reconstruction loss on the masked region. To show the advantages of the paper, the authors provide extensive comparisons with other baseline methods on several downstream tasks. The authors also conduct the ablation study to show the effectiveness empirically.

**Summary Of The Review:**

Overall, I do appreciate the messages that this paper sends to us. It reinforces the claim of ConvNext that convolution networks can be better than transformers. However, the proposed framework seems very engineering oritented.

---

> ### Author Response · Authors · 2022-11-06
> **Official Response to Reviewer Zhkz**
>
> We thank the Reviewer Zhkz for the constructive feedback; however, the reviewer's statements ("our work reinforces the claim of ConvNext that convnets can be better than ViTs", "sparse convolution has already solved the adaptation problem of convnet in BERT pre-training", etc.) seem to misunderstand the *core objectives and contributions* of our work, resulting in a misjudgment of the value and novelty. We regret for the misunderstanding, and would like to reiterate the keys below (which are taken from the Abstract and "To summarize..." part in Introduction of the original manuscript):
>
> - **The primary goal of this paper is to address the challenge that "BERT is still not successfully applied to convnets", not to discuss "convnets are better or ViTs better".**
>   - We have also concluded our work as "suggesting a promising future of BERT pre-training on convnets". In this regard, our goal is radically different from ConvNeXt's. The conclusions from ConvNeXt won't diminish our key contribution.
>
> - **Our key contribution is addressing the challenge above. Achieving this goal is important, and requires non-trivial efforts because:**
>   - The main contribution of existing ViT-based BERT-style pre-training (BEiT [r1], MAE [r2], SimMIM [r3], etc.) is they proved that "BERT works not only for *language transformers*, but also for *vision transformers*" (by using a large mask ratio of 0.4/0.6/0.75).
>   - However, since BERT is *naturally designed for transformers*, how to "extend BERT from *language or vision transformers* to *convnets*" is a huge and unresolved challenge. And our work is the **first** to solve it.
>   - **There have been efforts** over the past year trying to port BERT to convnets, yet eventually compromised on proposing a hybrid convolution-transformer model [r4] or non-masked modeling [r5], **which indicates the difficulty and our contribution/novelty**.
>   - One can therefore observe that our contributions and efforts are fundamentally different from existing work's (BEiT, MAE, SimMIM), and both are non-trivial. The reviewer doubts our novelty "the contribution to the community is much less than the published ones", which is somewhat excessive.
>
> - **Our novel use of sparse convolution**:
>   - Sparse convolution was invented and heavily used in 3D vision as a **pure** efficiency-oriented tool to reduce the computation complexity of **3D point cloud** processing.
>   - To the best of our knowledge, we are the **first** to observe the similarity between 3D point clouds and 2D masked images in BERT pre-training. And this is the **first** use of sparse convolution with the purpose of "facilitating the adaptation of convnet to masked images", rather than of "speeding up convolutions".
>   - Thus, the [Cons 1.a.] (details are in next) does not consider our contribution/novelty very properly.
>
> &nbsp;
>
> Despite the misunderstanding, we appreciate Reviewer Zhkz's comments as showing room for improvement in the clarity of our paper. We will update some of the statements in the paper accordingly. We then go on to answer the specific questions of Reviewer Zhkz:
>
> > [Cons 1.a.] Removing information of masked parts is not difficult for ConvNet. It has been solved by using submanifold sparse convolution.
>
> A: This is indeed our contribution. **We are the first** to propose to use submanifold sparseconv to solve this difficulty of convnet.
>
> > [Cons 1.b.] The work "ConvNext" has already shown convnets is not inferior to transformers.
> > [Cons 3.] In ablation study (Tab. 5), even without the hierarchical encoding, ConvNext outperforms the transformers.
>
> A: These facts won't devalue our main contribution of "successfully applying BERT to convnets".
>
> > [Cons 2.] In Fig. 1(c), why does the sparsely dropping not decrease the frequency of other pixels?
>
> A: Fig. 1 plots the probability distributions of pixel intensity. Random dropping only reduces the counts, not the probability. Therefore the probability distributions before and after the dropping are similar.
>
> > [Other doubts about originality/novelty]
>
> A: Please see our reiteration at the beginning of this response.
>
> &nbsp;
>
> We hope that the above response will help to resolve the confusion of Reviewer Zhkz.
> We are lookling forward to the reviewer's response.
>
> ---------------
>
> [r1] Bao, Hangbo, et al. "BEiT: BERT Pre-Training of Image Transformer." International Conference on Learning Representations. 2021.
>
> [r2] He, Kaiming, et al. "Masked autoencoders are scalable vision learners." Proceedings of the IEEE/CVF Conference on Computer Vision and Pattern Recognition. 2022.
>
> [r3] Xie, Zhenda, et al. "Simmim: A simple framework for masked image modeling." Proceedings of the IEEE/CVF Conference on Computer Vision and Pattern Recognition. 2022.
>
> [r4] Gao, Peng, et al. "ConvMAE: Masked Convolution Meets Masked Autoencoders." arXiv preprint arXiv:2205.03892.
>
> [r5] Fang, Yuxin, et al. "Corrupted image modeling for self-supervised visual pre-training." arXiv preprint arXiv:2202.03382.

---

> > ### Author Response · Authors · 2022-11-11
> > **Authors are kindly looking forward to your response**
> >
> > Thanks again for your time and efforts. Our clarifications on your queries are ready. We would appreciate it if you could take a look and get back to us if there are any further questions.

---

> > ### Comment · Reviewer_Zhkz · 2022-11-15
> > **Reply to the response**
> >
> > After reading the response from the authors, I have to say that I am not convinced by the rebuttal at all.
> >
> > 1. Following the story of what the authors suggested above, this paper's goal is "suggesting a promising future of BERT pre-training on convnets", however,  its essence is image inpainting to me. Please refer to the references below. Using the pretext task for self-supervised learning has been heavily studied before the MAE, even before the BERT.
> >
> > 2. On the technical side, "sparse convolution" seems not the only method to solve the problem. The authors can refer to the papers below to find a way. The very first paper of "sparse convolution" is used for 2D binary images or masks, I can not see why the authors claim the "first use" as a contribution.
> >
> > In summary, I don't think the novelty provided by the authors can convince me, but more importantly, they do not compare with other related techniques that can handle masks in convnets as I mentioned below.
> >
> >
> >
> >
> > [1] "Image Inpainting for Irregular Holes Using Partial Convolutions", Guilin Liu Fitsum A. Reda Kevin J. Shih Ting-Chun Wang Andrew Tao Bryan Catanzaro, ECCV, 2018
> >
> > [2] "Learning pyramid-context encoder network for high-quality image inpainting" Zeng, Yanhong and Fu, Jianlong and Chao, Hongyang and Guo, Baining, CVPR 2019
> >
> > [3]."Region-aware Adaptive Instance Normalization for Image Harmonization" Jun Ling, Han Xue, Li Song, Rong Xie, Xiao Gu,  CVPR 2021
> >
> > [4]. "Foreground-aware Image Inpainting" Wei Xiong, Jiahui Yu, Zhe Lin, Jimei Yang, Xin Lu, Connelly Barnes, Jiebo Luo, CVPR 2019.

---

> > > ### Author Response · Authors · 2022-11-17
> > > **Thank you, we have a further response and clarification**
> > >
> > > First, the reviewer said "not convinced by the rebuttal at all", but their latest questions are only about the Cons. 1 above. We would like to know if the reviewer has further concerns on our previous response to Cons. 2 and 3.
> > >
> > > As for the two follow-up concerns about Cons. 1, we find some of them seem wrong or misunderstand us. We'd like to address them as below:
> > >
> > > ---------------
> > > (i) &nbsp;Our main contribution is "**successfully** applying BERT(inpainting)-style self-supervised learning to convnets" (as in our first response to [Cons 1.b.]).
> > >
> > > > "Using the pretext task for self-supervised learning has been heavily studied before the MAE, even before the BERT."
> > >
> > > A: The existence of pioneering work [r6, r7] won't devalue our main contribution above.
> > > By "**successfully**", we mean SparK is the first to make BERT(inpainting)-style pre-training **far surpass** supervised pre-training on convnets, which **justifies** the pre-training's effectiveness. This is a non-trivial achievement because:
> > > 1. The pioneering work [r6, r7] initially explored inpainting pre-training for convnets but **performed far below** (more than $10$% worse) than supervised pre-training.
> > > 2. The recent efforts [r4, r5] have tried to copy the success from transformers to convnets, yet compromised on using a non-convolutional model or a non-inpainting(BERT) task for pre-training.
> > > 3. SparK can be directly used on any convnets and can outperform supervised pre-training by large margins (up to $3.5$% absolute improvements). **This leap in convnet's performance is desirable [r4, r5], but never achieved before**.
> > >
> > > ---------------
> > >
> > > (ii) &nbsp;For the relationship between [1-4] and our work:
> > >
> > > > "Please refer to [1-4]. Using the pretext task for self-supervised learning (SSL) has been heavily studied before the MAE, even before the BERT."
> > >
> > > A: [1-4] work only on **inpainting** task without the purpose of SSL. We thus discussed two **inpainting-motivated SSL** [r6, r7] in this response above.
> > >
> > > > "On the technical side, sparse convolution seems not the only method to solve the problem (can refer to [1-4] to find a way)."
> > > > "They do not compare with other related techniques that can handle masks in convnets, like [1-4]."
> > >
> > > A: We disagree with this because:
> > >
> > > 1. **In essence, [1-4] and sparse convolution have the exact opposite design and effect.** [1-4] use partial convolution (PConv) or its variants, whose primary goal (as claimed in [1]) is to make binary mask eventually be all ones with sufficient conv layers. However, this would raise the "mask pattern vanishing" issue (as shown in Figure 3 of our paper), which we exactly want to avoid (by using sparse convolution).
> > > 2. **We have also compared [1-4]-style pre-training (which raises the mask issue) with SparK**, and found it only achieved fine-tuning performance similar to non-pretraining (as in row 3 of Table 5). This further demonstrates the uniqueness and effectiveness of sparse convolution (SparK).
> > > 4. We have also done our best to check more variants ([r8, r9, r10], etc.), and found **no existing** method works the same as sparse convolution.
> > > 5. **[Another evidence of our novelty/contribution]:** We observe the fact that PConv works well in inpainting but fails in inpainting-motivated pre-training. This illustrates using advanced inpainting techniques [1-4] may not be beneficial (even harmful) for pre-training, while using the sparse convolution is the key to a successful pre-training.
> > > ---------------
> > >
> > > (iii) &nbsp;Our originality in "using submanifold sparse convolution for BERT-style self-supervised learning" can be evident by two facts:
> > >
> > > 1. There is no existing method that uses submanifold sparse convolution **for BERT(inpainting)-style self-supervised learning**.
> > > 2. The invention of submanifold sparseconv in 3D vision is known to be pure efficiency-driven. SparK first uses it with the purpose of "better self-supervised learning", rather than of "speeding up convolution". As also acknowledged by Reviewer 71p6, this is indeed a **new** understanding of submaifold sparseconv.
> > >
> > > &nbsp;
> > >
> > > We hope our further response and clarification could be helpful. Thanks again for your response :-).
> > >
> > > ------------
> > >
> > > [r6] Pathak, Deepak, et al. "Context encoders: Feature learning by inpainting." CVPR 2016.
> > >
> > > [r7] Zhang, Richard, Phillip Isola, and Alexei A. Efros. "Split-brain autoencoders: Unsupervised learning by cross-channel prediction." CVPR 2017.
> > >
> > > [r8] Lee, Cheng-Han, et al. "Maskgan: Towards diverse and interactive facial image manipulation." CVPR 2020.
> > >
> > > [r9] Yu, Jiahui, et al. "Free-form image inpainting with gated convolution." ICCV 2019.
> > >
> > > [r10] Li, Jingyuan, et al. "Recurrent feature reasoning for image inpainting." CVPR 2020.

---

### Author Response · Authors · 2022-12-13
**[Author Rebuttal Summary] on Dec. 12, 2022**

We sincerely thank all reviewers and chairs for their comments and responses.
We appreciate it a lot that our contributions and novelty have been acknowledged by reviewers:


- **[main contribution]** "transferring the successful experiences of MIM from transformers to convnets" (Reviewer 71p6)

- **[novelty: the first successful solution]** "provides a pioneering solution" (Reviewer 71p6)

- **[novelty: the first use of sparseconv in MIM/BERT]** "the idea of using the submanifold sparse convolution to handle irregular masked input is well-motivated and new" (Reviewer 71p6)

- **[effectiveness]** "simple and experimentally validated", "strong results", "detailed ablation study" (Reviewer tYbJ)


&nbsp;

-------------


During the rebuttal, we made every effort to address the concerns of reviewers, and we appreciate if these improvements could be considered. All revisions are highlighted in blue in our updated manuscript. Main modifications are:

- adequate discussions on related work (Reviewer Zhkz, 71p6)

- more results of baselines and heavier pretraining (Reviewer 71p6, tYbJ)

- useful empirical results including more ablations, reconstruction losses, error bars/experiment variance, and linear evaluation results (Reviewer tYbJ).

- descriptions about backgrounds, motivations, and efficiency comparison (Reviewer 71p6)


&nbsp;

-------------


Thanks again for everyone's time and efforts. We value every feedback a lot, and would like to discuss more in the rest phase and in the future.

Best,

Paper 113 authors

---

### Decision · Program_Chairs · 2023-01-20

**Decision:**

Accept: notable-top-25%

**Justification For Why Not Higher Score:**

The results are nice, but not that much better than baselines

**Justification For Why Not Lower Score:**

The paper has a clear message and supports it convincingly with extensive experiments. The fact that masked modeling works for convolutional models can be of general interest, so spotlight may be justified.

**Metareview: Summary, Strengths And Weaknesses:**

The paper proposes an approach for masked image modeling (MIM) pre-training with ConvNets, based on sparse convolutions. The proposed method is validated on multiple tasks and model architectures and shows strong results compared to relevant baselines.

The opinions of the reviewers about the paper are somewhat divergent even after the authors' rebuttal and discussion. Based on the reviews, rebuttals, and the paper itself, the main points are as follows:

Pros:
1. New and reasonable technical approach
2. Clear presentation
3. Extensive experiments on several tasks and model architectures, showing strong results compared to relevant baselines. A good ablation study.

Cons:
1. No results for very large models
2. Doubts about wallclock computational efficiency
3. Inpainting has been done before

Con (1) is understandable since training those very large models takes an excessive amount of computational resources. Con (3) in my opinion is not really a problem - yes inpainting is not a new task, but self-supervised pre-training results using inpainting have been quite weak so far and there are no strong baselines there to compare to. Regarding con (2) - indeed would be nice to clearly report wallclock training/inference times for sparse vs usual convolution.

Overall, the paper has a clear message and supports it convincingly with extensive experiments. Therefore, I recommend acceptance.

**Note From Pc:**

if the above contains the word "oral" or "spotlight" please see: "oral" presentation means -> notable-top-5% and "spotlight" means -> notable-top-25%. As stated in our emails, we are disassociating presentation type from AC recommendations